# Study on Accuracy Evaluation of MCD19A2 and Spatiotemporal Distribution of AOD in Arid Zones of Central Asia

Zhengnan Zhu [1,2,3], Zhe Zhang [1,2,3,4,*], Fangqing Liu [1,2,3], Zewei Chen [1,2,3], Yuxin Ren [1,2,3] and Qingfu Guo [1,2,3]

1. College of Geography and Remote Sensing Sciences, Xinjiang University, Urumqi 830046, China; 107552101079@stu.xju.edu.cn (Z.Z.); liufangqing_0215@yeah.net (F.L.); 107552201136@stu.xju.edu.cn (Z.C.); 107552201161@stu.xju.edu.cn (Y.R.); 107552203688@stu.xju.edu.cn (Q.G.)
2. Xinjiang Key Laboratory of Oasis Ecology, Xinjiang University, Urumqi 830046, China
3. Key Laboratory of Smart City and Environment Modelling of Higher Education Institute, Xinjiang University, Urumqi 830046, China
4. MNR Technology Innovation Center for Central Asia Geo-Information Exploitation and Utilization, Urumqi 830046, China
* Correspondence: zhangzhe_0110@yeah.net; Tel.: +86-187-9918-1249

**Abstract:** The Central Asian arid zone is the largest non-territorial arid zone in the world, so it is particularly important to understand the optical properties of aerosols in this region. In this paper, we validate the MCD19A2 atmospheric aerosol optical depth (AOD) remote sensing data by using ground-based data and measured data. To explore the spatial and temporal changes in aerosols in the Central Asian arid zone as well as the interannual variations and seasonal variations, we characterize the spatial and temporal distributions of the AOD over 20 years. Finally, we analyze the spatial and temporal variations of the AOD in the Central Asian arid zone by using three methods, namely, the Theil–Sen median trend analysis combined with the Mann–Kendall test, coefficient of variation, and Hurst index; analyze the characteristics of the spatial and temporal variations of the AOD in the Central Asian arid zone; and explore the relationships among the AOD, wind speed, and NDVI. This study reveals the characteristics of the long-term changes in the aerosol optical properties in the Central Asian arid zone and provides a scientific basis for estimating the factors affecting climate change.

**Keywords:** aerosol; MCD19A2 validation; spatial and temporal variations

## 1. Introduction

Aerosol is a general term for suspended solid–liquid particles in the atmosphere with diameters of 0.001~100 μm, which can directly or indirectly affect solar radiation through absorption, scattering, and other processes and, thus, change the energy balance of the ground–air system. Meanwhile, aerosols, as an important type of cloud condensation nuclei, have an important impact on local precipitation by participating in cloud microphysical processes. In addition, atmospheric pollution caused by aerosols seriously affects human health [1,2]. The aerosol optical depth (AOD), as an important parameter of aerosol optical properties, is the integral of the extinction coefficient of the aerosol between the top of the atmosphere and the ground, which indicates the degree of reduction of the light that is scattered and absorbed by the aerosol in the plumb column in a cloudless sky and can reflect the aerosol content. The study of aerosol optical properties based on the AOD provides an important data reference for the quantitative assessment of the degree of air pollution, which is of great significance for determining the degree and type of air pollution and for understanding the characteristics of aerosols and their changes in the arid zones of Central Asia.

Aerosols show significant spatial heterogeneity due to their characteristics, subsurface, and atmospheric environment, and many scholars at home and abroad obtained the spatial

and temporal distribution patterns of air pollution at the regional scale through in-depth studies of the optical characteristics of aerosols such as the AOD, Ångström wavelength index, and single-scattering albedo, which provide an important guideline for the prevention, control, and management of atmospheric pollution. In traditional AOD study, researchers analyzed the spatial and temporal distributions of the AOD by using ground observation data such as solar photometers, ground-based monitoring stations, etc. Semenov analyzed the spatial and temporal distributions of the AOD in the Issyk-Kul region by using the measured data from a Microtops II [3]; the local atmospheric column aerosol characteristics obtained by Rupakheti et al. from the long-time-scale aerosol data of the Issyk-Kul region were obtained by a Cimel solar photometer [4]. Traditional ground-based monitoring can obtain high-precision data for subsequent research, but its inability to carry out long-time-series monitoring and the uneven distribution of stations, ignoring the spatial heterogeneity of aerosols and other inherent shortcomings, limit the further development of aerosol monitoring; while the rapid development of remote sensing technology solves the problem, satellite remote sensing breaks through the spatial and geographic constraints, facilitating the acquisition of a wide range of long-time-series changes in atmospheric aerosols, making up for the shortcomings of the spatial scales in ground-based monitoring [5]. At the same time, more remote sensing products applicable to atmospheric research are constantly appearing, and there are deeper explorations into aerosol characteristics, aerosol types, the validation of the universality of satellite data, etc. Scholars at home and abroad discussed the aerosol optical characteristics and spatial and temporal distribution patterns in different regions through different aerosol optical parameters, such as the AOD, wavelength index, turbidity coefficient, and single-scattered albedo, to reveal the characteristics of the changes in long time series [6–10]. Niu et al. explored the spatial and temporal distribution characteristics of the AOD's long time series in five countries of Central Asia by analyzing the base station observation data [11]; Zhang et al. analyzed that the changes in the AOD were affected by a combination of meteorological factors (wind speed, air temperature, relative humidity, etc.) [12]; J. Liu et al. extracted and calculated the daily sand and dust aerosol optical thickness data in Central Asia for the period of 2003–2018 and analyzed the trend of their change [13]; Falah et al. investigated the performance of the MAIAC AOD inversion under surface reflectance/land cover (based on the normalized difference vegetation index (NDVI)) [14]. Accurately grasping the characteristics of aerosols is important for the mitigation of atmospheric pollution as well as ecological construction, but given the formation of aerosols by their subsurface and local environmental coupling, the characteristics of aerosols in different regions have obvious differences; in the analysis of the characteristics of aerosols in a particular place, we need to take into account the characteristics of the local environment to grasp the characteristics of aerosols [15–17].

Central Asia, which is located in the water transition region, is a typical arid and semi-arid region, with a sensitive climate, a fragile ecosystem, harsh natural conditions, and a severe lack of local water resources and vegetation, and is also the region that contributes the most to global sand and dust aerosols [12]. This region is characterized by significant seasonal variations, and dust dispersion in the Aral Sea region exhibits significant longitudinal and latitudinal dispersion characteristics [18]. At the same time, the serious desertification of the lakeshore, as a result of the drying up of the Aral Sea, is a matter of concern for scholars in many countries [19]. However, due to the harsh natural conditions and local economic development level, there are fewer ground-based aerosol observation stations in the Central Asian arid zone, and there are not many studies on the optical properties of aerosols in the Central Asian arid zone, mainly around the Aral Sea and in the Tarim Basin, over a long time series.

In this paper, we use ground-based solar photometer observation data to verify the accuracy of the AOD product (MCD19A2) of MODIS satellite remote sensing data on a long time scale. The aims of the study were to characterize the spatial and temporal distribution of AOD in the Central Asian arid zone over the 20 years from 2001 to 2020, to discuss the seasonal differences, and to analyze the spatial characteristics of the AOD in the Central

Asian arid zone on the metric scale, the degree of stabilization, and the development trend in the future. To accurately grasp the optical characteristics of aerosols in the Central Asian arid zone, we will provide a theoretical foundation and scientific basis for the prevention and control of air pollution and the management of sand and dust sources in the region.

## 2. Materials and Methods

### 2.1. Overview of the Study Area

The arid zone of Central Asia refers to the five Central Asian countries (Kazakhstan, Turkmenistan, Uzbekistan, Tajikistan, and Kyrgyzstan) and the Xinjiang Uygur Autonomous Region (XUAR) of China, which has a high degree of ecological similarity at the same latitude. It is located between 35°–57° N and 48°–96° E, with a total area of $5.6 \times 10^6$ km$^2$. In Figure 1, Kazakhstan is abbreviated as Kaz, Turkmenistan is abbreviated as Tkm, Uzbekistan is abbreviated as Uzb, Tajikistan is abbreviated as Tjk, Kyrgyzstan is abbreviated as Kyr, the Xinjiang Uygur Autonomous Region (XUAR) of China is abbreviated as XJ. Influenced by the Tien Shan Mountains, the Kunlun Mountains, and the Pamir Plateau, the topography is high in the southeast and low in the northwest, and at the same time, the region is the largest non-zonal arid zone in the world, which is mainly controlled by the mid-latitude westerly circulation, and has a significantly different climatic and environmental characteristics than the Asian monsoon zone. The Asian monsoon zone is significantly different, resulting in a mostly temperate continental climate [20], with cold winters and hot summers, scarce water vapor, and an extremely sensitive and fragile ecosystem. The arid zone of Central Asia is the core region for the implementation of the Belt and Road Initiative, and with the vigorous promotion of economy, industrialization, and urbanization, the emission of pollutants into the atmosphere has increased significantly, and the air quality has been deteriorating. Therefore, understanding the local aerosol changes is of great significance to the prevention of air pollution and the management of dust and sand sources in the arid zone of Central Asia.

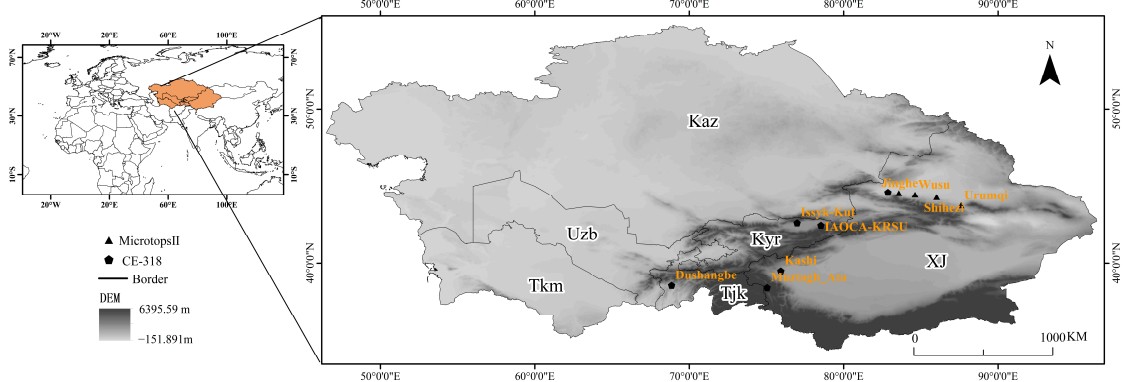

**Figure 1.** An overview map of the study area.

### 2.2. Data Presentation

#### 2.2.1. MODIS MAIAC AOD Data

MAIAC AOD (MCD19A2) data are a level 2 product of land AOD gridded based on the new advanced MAIAC algorithm [5,21]. In this paper, the AOD data product MCD19A2 generated based on the MAIAC algorithm is selected, which is a combination product based on Terra/Aqua with 1 km spatial and temporal resolution, and the data can be downloaded from the NASA official website (https://ladsweb.modaps.eosdis.nasa.gov/search/order (accessed on 1 March 2023)). The MAIAC algorithm decouples aerosol and surface contributions using time-series feature data, assuming stable and spatially heterogeneous subsurface conditions over short periods. At the same time, the product analyzes bi-directional surface reflectance, and the combination of time scale and spatial deconvolution greatly improves the quality of cloud and snow detection [22]. The MAIAC

products include cloud masks, 0.47 and 0.55 μm AOD, along with AOD-related quality control documentation and uncertainty data. For data processing in this study, MCD19A2 data covering the period January 2001–December 2020 were selected, and the dataset was preprocessed to extract the AOD values at 550 nm by using the ENVI/IDL programming tool, which was used as a basis to obtain the distribution of the mean AOD values at different time scales for each month, season, and year in the last 20 years. Meanwhile, to avoid the chance of changes in the mean AOD between seasons in different years, March–May was classified as spring, June–August as summer, September–November as fall, and December–February as winter according to the meteorological division standard.

### 2.2.2. Solar Photometer Data

A total of two types of sun photometer data were used in this study to assist in the validation of the MAIAC AOD product. The first one is the CE-318 sun photometer and the second one is the Microtops II sun photometer. The observation sites selected in this paper are Muztagh_Ata, Dushanbe, Issyk-Kul, IAOCA-KRSU, JingheI, and Kashi. The CE-318 solar photometer is a kind of automatic tracking and scanning solar photometer produced by the French company Cimel, which is the most widely used and authoritative solar photometer instrument today and is widely used in various aerosol monitoring network platforms, such as the US global AERONET, French PHOTONS, China Meteorological Administration, and China Oceanic Administration. It can reflect the optical properties of the atmosphere by observing the Sun, sky, and ground reflections. The instrument has nine spectral channels in the visible near-infrared band and can acquire AOD values at 340, 380, 440, 500, 675, 870, 936, 1020, and 1640 nm. The results of data processing of the CE-318 solar photometer consist of two quality levels: level 1.5 data are the data without de-clouding and level 2.0 data are the data with de-clouding, and level 2.0 data were chosen for the present study. In this study, the inversion data of level 2.0 were acquired through the AERONET ground-based aerosol observation network (https://aeronet.gsfc.nasa.gov (accessed on 1 March 2023)). Microtops II is a portable, handheld, multi-band sun photometer that can measure direct solar radiation and AOD data in the corresponding wavelength bands. Microtops II can select five channels with a field of view of 2.5° and can accurately measure the aerosol properties of the selected channels and obtain high-precision aerosol data for the corresponding channels. The instrument is widely used in aerosol optical properties research. The instrument has nine spectral channels in the visible near-infrared (NIR) band and can acquire AOD values at 340, 380, 440, 500, 675, 870, 936, 1020, and 1640 nm. The observation sites, shown in Figure 1, are located in Jinghe County and the cities of Wusu, Shihezi, and Urumqi. To avoid the interference of human activities in the aerosol information in the environment as much as possible, most of the observation sites were located in the open area upwind of the main area without obvious pollution sources, and at the same time, to avoid the influence of anthropogenic aerosols caused by ore burning, the observation time was mainly concentrated in the non-heating period from April to October, and the time of the observation was from 09:30 to 21:00 Beijing time, with a sampling interval of 15 min, and each observation was consecutive. Three measurements were taken; the absolute error of the three sets of data did not exceed 0.03 to define a valid observation.

### 2.2.3. Meteorological Data and Vegetation Normalization Data

The source of meteorological data is the ERA5 monthly mean reanalysis data. ERA5 is the fifth generation of ECMWF reanalysis data of the European Centre for Medium-Range Weather Forecasts (ECMWF) for Global Climate and Weather, and the main meteorological factor used in this paper is wind speed. The spatial resolution is 0.25° × 0.25°, and the temporal resolution is 1 m. The vegetation index product selected for this paper is the MOD13A2 normalized vegetation index product describing the ground vegetation condition, with a spatial resolution of 1 km and a temporal resolution of 16d.

*2.3. Methods*

2.3.1. Validation Methods

To validate the reliability of MCD19A2 data in the Central Asian arid zone, ten stations were selected (Table 1), Muztagh_Ata, Dushanbe, Issyk-Kul, IAOCA-KRSU, Jinghe I, Kashi, Urumqi, Jinghe II, Shihezi, and Wusu, for validation of the MCD19A2 data, of which the first six stations are CE-318 sun photometer stations and the last four are Microtops II sun photometer stations.

**Table 1.** Solar photometer station information.

| Site | Longitude | Latitude | Observation Time | Sun Photometer |
|---|---|---|---|---|
| Muztagh_Ata | 75.039 | 38.408 | June 2011–November 2011 | CE-318 |
| Issyk-Kul | 76.983 | 42.623 | August 2007–May 2016 | CE-318 |
| IAOCA-KRSU | 78.529 | 42.464 | March 2014–May 2017 | CE-318 |
| Dushanbe | 68.858 | 38.553 | July 2010–April 2018 | CE-318 |
| Kashi | 75.9304 | 39.5043 | March 2019–April 2019 | CE-318 |
| JingheI | 82.864 | 44.599 | June 2019–December 2019 | CE-318 |
| Urumqi | 87.600 | 43.750 | October 2018–July 2019 | Microtops II |
| Shihezi | 86.017 | 44.294 | May 2019–June 2019 | Microtops II |
| Wusu | 84.620 | 44.450 | July 2015–September 2018 | Microtops II |
| JingheII | 83.568 | 44.545 | July 2015–August 2018 | Microtops II |

Based on the inversion of the MAIAC algorithm, the AOD value at 550 nm is obtained. To compare and analyze the sun photometer data with MODIS remote sensing AOD products, and to better match the AOD values of the two products in the corresponding wavelength bands and spatial and temporal scales, the AOD value at 550 nm is obtained from the sun photometer data by using the Ångström exponential formula. Using AOD data at wavelengths closer to 550 nm, typically 675 nm and 440 nm, the AOD at 550 nm was calculated for the sun photometer data [23]. According to Ångström's formula:

$$\text{AOD}_{550\text{nm}} = \text{AOD}_{440\text{nm}}(550/440)^{-\alpha} \tag{1}$$

where $\alpha$ is Ångström's wavelength index at 440–675 nm.

The observation period of the sun photometer data is 15 min, while the MCD19A2 data are the AOD values during the satellite transit period, so it is necessary to match the sun photometer data with the MCD19A data in time and space. On the time scale, based on the satellite transit period, the sun photometer data within 30 min of the period are selected and averaged. In contrast, for the spatial matching, the satellite retrieval within the 550 km diameter of the CE-318 sun photometer site is used to obtain the effective AOD averages [24,25]. Also in this study, three statistical parameters, sample correlation coefficient (R), root mean square error (RMSE), and mean absolute error (MAE), were used to evaluate the precision of MCD19A2 data.

Mean absolute error (MAE) [26] is a measure of the difference between two continuous variables and is calculated as follows:

$$\text{MAE} = \frac{1}{n}\sum_{i=1}^{n}\left|\tau_{(\text{MODIS})i} - \tau_{(\text{AERONET})i}\right| \tag{2}$$

The root mean square error (RMSE) is sensitive to both systematic and random errors, so it was utilized in this study to measure the difference between the MCD19A2 data and the sun photometer data. The RMSE was calculated using the following formula:

$$\text{RMSE} = \sqrt{\frac{1}{n}\sum_{i=1}^{n}\left(\tau(\text{MODIS})_i - \tau(\text{AERONET})_i^2\right)} \tag{3}$$

The number of sample points within the expected error (EE) also reflects the quality of the MCD19A2 data and is calculated as follows:

$$EE = \pm(0.05 + 0.2AOD_{AERONET})$$ (4)

When the inversion result of MCD19A2 data matches AOD foundation $-EE \leq AOD$ MODIS $\leq$ AOD foundation $+EE$, it indicates that the inversion result of MCD19A2 data is better, which can reflect the spatial and temporal distribution of AOD better. However, due to the limited observation time and fewer matching samples, the evaluation is only for preliminary reference, and a longer time series of ground-based observation data is needed for better evaluation and validation.

2.3.2. Trend Analysis

The combination of Theil–Sen median trend analysis [27] and Mann–Kendall test methods [28] can simulate the trend of each raster and comprehensively characterize the evolution of regional patterns of a certain time series through the spatial change characteristics of individual image elements at different times. The Theil–Sen median method, also known as the Sen slope estimation, is a non-parametric statistical method of trend calculation. Because the method is computationally efficient, insensitive to outlier data and measurement errors, and has a strong resistance to data errors, it can objectively reflect the evolutionary trend of the long-time-series AOD, which is calculated as follows:

$$S_{AOD} = \text{Median}\left(\frac{AOD_j - AOD_i}{j - i}\right) \quad 2001 \leq i \leq j \leq 2000$$ (5)

where $S_{AOD}$ denotes the slope of the one-dimensional linear fitting equation; $AOD_i$ is the value of AOD in year $i$; and $AOD_j$ is the AOD in year $j$. $S_{AOD} > 0$ indicates an increasing trend in the change in AOD, and vice versa indicates a decreasing trend in AOD.

The Mann–Kendall significance test method, also known as the M-K test, is used to test the significance of the slope of the data over a long time series, i.e., the Sen trend value is first calculated, and then the M-K is used to determine the significance of the trend. Both this method and the Sen slope estimation method do not require the sample to follow a normal distribution, so the results are not easily affected by outliers. The method is used to calculate the spatial variation characteristics of AOD data on a pixel-by-pixel basis.

The calculation formula is as follows: setting $AOD_i$, $I = 2001, 2002, \ldots, 2020$.

Set the parameter Z as:

$$Z = \begin{cases} \frac{S-1}{\sqrt{s(S)}}, S > 0 \\ 0, S = 0 \\ \frac{S+1}{\sqrt{s(S)}}, S < 0 \end{cases}, S = \sum_{j=1}^{n-1}\sum_{i=j+1}^{n} sgn(AOD_i - AOD_j)$$ (6)

$$sgn(AOD_i - AOD_j) = \begin{cases} 1, AOD_j - AOD_i > 0 \\ 0, AOD_j - AOD_i = 0 \\ -1, AOD_j - AOD_i < 0 \end{cases}, s(S) = \frac{n(n-1)(2n+5)}{18}$$ (7)

where $AOD_i$ and $AOD_j$ denote the AOD values in the year $i$ and $j$ of the image element, respectively; $n$ denotes the length of the time series; $sgn$ is the sign function; and the value of the statistic Z ranges from $(-\infty, +\infty)$. At a given level of significance, when $Z > u_{1-\alpha/2}$, it indicates that there is a significant change in the study series at the $\alpha$ level. In this paper, we determine the significance of the trend of change in the AOD time series at the $\alpha = 0.05$ confidence level. Normally, taking $\alpha$ as a 0.05 confidence level, when the absolute value of Z is greater than 1.28, 1.96, and 2.32, its distribution passes the significance test with confidence levels of 90%, 95%, and 99%.

Regarding the trend of AOD changes over the twenty years, a combination of Theil–Sen median trend analysis and Mann–Kendall test results with $Z \geq 1.96$ passed the significance

test with a confidence level of 95%. The interannual trends of aerosol changes were categorized into five classes (Table 2).

**Table 2.** Grading criteria for AOD changes in arid zones, 2001–2020.

| $S_{AOD}$ | Z-Value | Changes in AOD Trends |
|---|---|---|
| $\geq 0.05$ | $\geq 1.96$ | increase significantly |
| $\geq 0.05$ | $-1.96 \sim 1.96$ | slight increase |
| $-0.05 \sim 0.05$ | $-1.96 \sim 1.96$ | unchanged |
| $< -0.05$ | $-1.96 \sim 1.96$ | slight decrease |
| $< -0.05$ | $< -1.96$ | significant reduction |

### 2.3.3. Coefficient of Variation

The coefficient of variation has no scale, reflecting the overall dispersion of the observed values, and the larger the value, the larger the difference and the more obvious the relative change, which is generally applicable to the case where the mean value is not zero. The coefficient of variation is used to calculate the stability of each raster AOD year by year from 2001 to 2020, to reveal the characteristics of intrayear changes in AOD. The coefficient of variation, a measure of the dispersion of the data distribution, is used to assess the stability of the AOD over the time series and is calculated by the following formula:

$$Cv = \frac{\sigma}{\mu} \tag{8}$$

where $Cv$ is the coefficient of variation, $\sigma$ and $\mu$ are the standard deviation and mean of the observations, $Cv$ is dimensionless to reflect the overall dispersion of the observations; the smaller the value of $Cv$, the smaller the fluctuation of the time series and the better the stability, and it applies to the case where the mean value is not zero.

### 2.3.4. The Hurst Index

The Hurst index quantitatively describes the persistence of the time series, which has a wide range of applications in the research fields of climatology and geochemistry, etc. The Hurst index based on the rescaled polar analysis method can predict the future development trend of a specific spatiotemporal series [29], which is used to analyze the persistence characteristics of AOD. For the time series, $t = 1, 2, \ldots, n$, define the mean series and the calculation formula is shown as follows.

$$\text{AOD}_{(\tau)} = \frac{1}{\tau} \sum_{t=1}^{\tau} \text{AOD}_{(\tau)} \quad \tau = 1, 2 \ldots n \tag{9}$$

$$\text{cumulative deviation}: \ X_{(t,\tau)} = \sum_{t=1}^{t} \left( \text{AOD}_{(t)} - \text{AOD}_{(\tau)} \right) \ 1 \leq t \leq \tau \tag{10}$$

$$\text{extremely poor}: \ R_{(\tau)} = maxX_{(t,\tau)} - minX_{(t,\tau)} \ \tau = 1, 2, \cdots, n \tag{11}$$

$$\text{standard deviation}: \ S_{(\tau)} = \left[ \frac{1}{\tau} \sum_{t=1}^{\tau} \left( \text{AOD}_{(t)} - \text{AOD}_{(\tau)} \right)^2 \right]^{\frac{1}{2}} \ \tau = 1, 2, \cdots, n \tag{12}$$

For the ratio $R_{(\tau)}/S_{(\tau)} \triangleq R/S$, if there exists the relationship $R/S \propto \tau^H$, it indicates that there is a Hurst phenomenon in the time period under study, where $H$ is the Hurst index. The value range of Hurst is from 0–1; if $0.5 < H < 1$, it indicates that the time series under study is persistent, and the closer the H value is to 1, the more persistent it is; if $H = 0.5$, it indicates that the AOD time series is stochastic; if $0 < H < 0.5$, it indicates that the time series under study has antipersistence, and the closer the value is to 0, the stronger the antipersistence is.

### 2.3.5. Correlation Analysis

In this paper, we analyze the correlation of AOD with wind speed and NDVI using a pixel-based spatial approach [30]. The formula is as follows:

$$R_{x,y} = \frac{\sum_{i=1}^{n} (x_i - \overline{x})(y_i - \overline{y})}{\sqrt{\sum_{i=1}^{n} (x_i - \overline{x})^2 \sum_{i=1}^{n} (y_i - \overline{y})^2}} \tag{13}$$

$R_{x,y}$ is the correlation between the variable $x$ and the variable $y$; $x_i$ is the AOD for a year $i$; $y_i$ is the value of another NDVI variable, wind speed, in year $i$; $\overline{x}$ is the average AOD over the study period; $\overline{y}$ is the average of the other variables over the study period. We used a *t*-test [31] to determine significance. The results were categorized into five classes: $p < 0.05$, $R < 0$ for significant negative correlation; $p < 0.05$, $R > 0$ for significant positive correlation; $p > 0.05$, $R < 0$ for non-significant negative correlation; $p > 0.05$, $R > 0$ for non-significant positive correlation; and $R = 0$ for non-significance with missing data.

## 3. Results

### 3.1. MCD19A2 Applicability Assessment

With its high spatial and temporal resolution, the MCD19A2 data provide a finer spatial scale and time scale for quantitative studies of aerosol properties, classification, and loading. To verify its applicability in the Central Asian dry zone, three statistical parameters, namely, sample correlation coefficient (R), root mean square error (RMSE), and mean absolute error (MAE), were used to evaluate the applicability of MCD19A2 data in the Central Asian dry zone based on the measured data from the solar photometer and MCD19A2 data.

As can be seen in Figure 2, the overall effect of the two fits is good, where R is greater than 0.5, RMSE is located at 0.1 and below, and more than 50% of the validation data at all sites are within the expected error, which is in line with the EE range. The Muztagh_Ata site (Figure 2a) has some underestimation of high values, which is related to its high altitude; whereas the IAOCA-KRSU and Issyk-Kul sites are overestimated due to the sensitivity of surface albedo to surface salinity, which leads to the errors in the calculation of the surface radiative contribution; Dushanbe, Kashi, Urumqi, and Wusu have some underestimation, which may be because the solar photometer setup sites are more sensitive to the surface salinity. Dushanbe, Kashi, and Urumqi all showed some underestimation phenomena, which may be related to the fact that the solar photometer sites are located near the cities and are polluted by the urban areas; whereas Issyk-Kul and Jinghe I showed some overestimation tendency, which is because the sites are located near the rivers, the surfaces of the dry river beds are heavily salted, and the salts fall on the soil and thus have a certain effect on the albedo values, which is in line with related studies [24,25]. Shihezi and Jinghe II fitted better overall. Combined with the above description, MCD19A2 has better inversion results at representative sites in the arid zone, indicating that MODIS products are more accurate in the study of aerosol optical properties in the arid zone. Of course, there are some shortcomings in this study, mainly due to the large study area in the arid zone and the limited number of ground-based and measured sites. For validating MCD19A2, more sites are needed to support its accuracy.

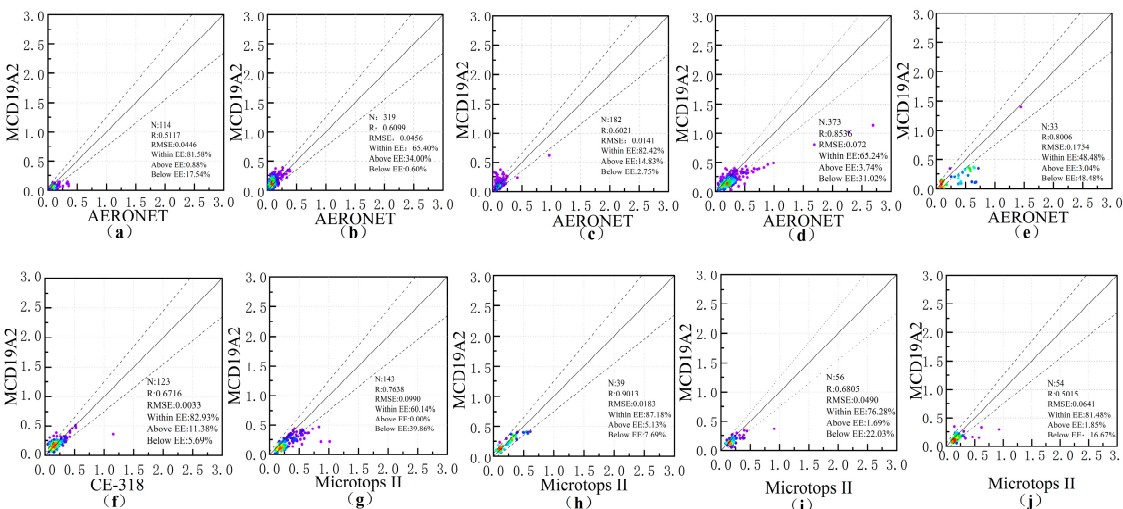

**Figure 2.** Comparison of MCD19A2 data with solar photometer data observations (where (**a**–**j**) are Muztagh_Ata, Issyk-Kul, IAOCA-KRSU, Dushanbe, Kashi, Jinghe I, Urumqi, Shihezi, Wusu, Jinghe II).

### 3.2. Aerosol Spatial and Temporal Distribution

The study yielded a map of spatial patterns of mean AOD values over a 20-year period in the arid zone of Central Asia (Figure 3a), while the multi-year mean value of 0.155 was calculated for the Central Asian arid zone, and the values of AOD in each region are shown in Table 3. From Figure 3a, under the influence of climatic conditions and subsurface factors, the distribution of AOD in the Central Asian arid zone shows obvious spatial heterogeneity, and with the increase in elevation, AOD shows a decreasing trend, presenting an obvious regular spatial distribution. The high values of AOD are mainly distributed in the deserts, basins, and lowlands of the Central Asian arid zone, such as the Taklamakan Desert, the lowlands along the Caspian Sea, and the Ferghana Basin, and the Aral Sea area also has relatively high values of AOD. Among them, the Tarim Basin in Xinjiang, China, has higher AOD values due to the local climatic factors and low ground vegetation cover, more sand and dust aerosols, and a lower altitude of the basin, where the aerosols gather and are not easy to diffuse, resulting in a higher AOD load, with a multi-year average value of 0.2027. The Ferghana Basin has similar values to the Tarim Basin. The Aral Sea region is also an area of high AOD value, which is due to the overexploitation and utilization of water resources in the region, global warming, and other factors, resulting in the rapid drying up of the Aral Sea and the rapid decline of the water level, exposing a large area of the lake bed, which increases the emergence of sand, dust, and salt-dust aerosols. At the same time, the Alarqum Desert in the Aral Sea region is the main source of dust in Central Asia, and the barren land area is one of the main sources of aerosols, releasing a large amount of salt-containing mineral dust, which is a major source of dust which is then transported to East Asia by strong westerly wind jets, and the wind–dust phenomenon is particularly significant in the Aral Sea. The high AOD in the lowlands along the Caspian Sea is because the Caspian Sea is a saltwater lake, the soil around the Caspian Sea is prone to salinization, and the local atmospheric concentration of salt dust is higher than in other regions under the influence of the westerly wind belt. In general, the northern part of the arid zone is flat, with low AOD values, a slight decrease in the trend, poor stability, and a weak inverse persistence in the future trend, which indicates that this area is more affected by natural influences and that there is a need to increase the measures to prevent and control AOD pollution. The central part of the arid zone has higher terrain and lower AOD values except for the Aral Sea, the trend of change increases, stability is better, and the future trend mainly shows weak persistence, indicating that anthropogenic factors are dominant. The southern part of the arid zone is mainly dominated by the Tarim Basin, which has low elevation, high AOD values, basically unchanged trend, and poor stability, and the future trend mainly shows weak anticontinuance, which indicates that the area has

more natural influences and needs to manage the surrounding environment and reduce the number of sand and dust days.

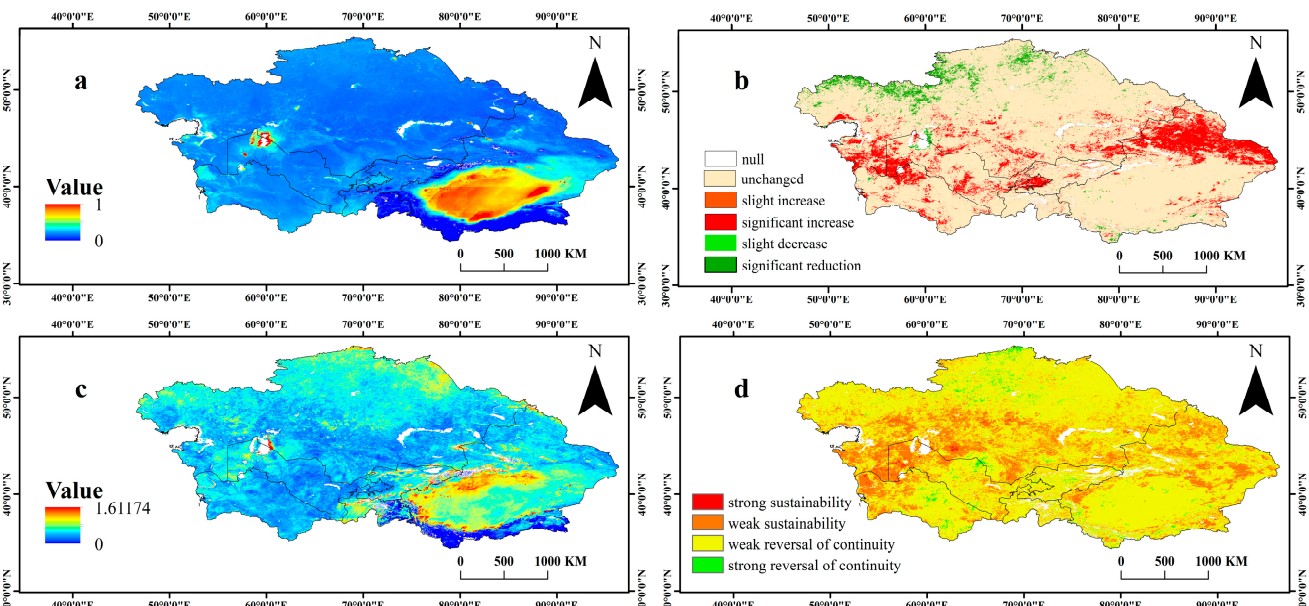

**Figure 3.** (**a**) The distribution of mean AOD values in the Central Asian arid zone during the 20 years, (**b**) the trend of the 20-year average annual AOD change, (**c**) the trend of AOD stability in 20 years, and (**d**) the trend of future changes in the arid zone of Central Asia.

**Table 3.** Values of AOD in the Central Asian arid zone for the period 2001–2020.

| Regions | Min | Max | Average |
| --- | --- | --- | --- |
| Tajikistan | 0.014 | 0.3824 | 0.0964 |
| Turkmenistan | 0.0753 | 0.4755 | 0.1462 |
| Uzbekistan | 0.0181 | 2.5329 | 0.1504 |
| Kazakhstan | 0.01399 | 2.4246 | 0.1258 |
| Kyrgyzstan | 0.106 | 0.1378 | 0.11923 |
| Xinjiang, China | 0.0139 | 1.4131 | 0.2027 |

From the trend map of interannual changes in AOD characteristics in the arid zone from 2001 to 2020 (Figure 3b), it can be seen that the spatial distribution of AOD changes in Central Asia during the 20 years showed obvious regional differences, with a small portion of the northern part showing a significant decreasing trend (0.3287%), while the AOD values in the central-western and central-eastern regions were significantly increased (13.6364%). The rest of the majority of the regions remained unchanged (83.0076%). As the core area of the Silk Road Economic Belt, the rapid development of the economy and urbanization in this region has brought about increasingly serious air pollution problems. Meanwhile, as a region downwind of the Alashankou, the northwestern monsoon winds can blow a large amount of dust and salt aerosols from Central Asia to this region, which leads to a significant increase in AOD in this region. Secondly, the Üstiert plateau on the east coast of the Caspian Sea also shows a large area of significant increase, which may be related to the partial drying up of the Caspian Sea, which leads to sand and dust aerosolization of the lake bed sediment, as well as the high altitude of this area and its location near the lake, which may be related to the overestimation of the MCD19A2 data for the area of the higher elevation and near the lake, as discussed in Section 3.1.

The coefficient of variation (CV) was calculated image by image from 2001 to 2020 to measure the interannual stability of the AOD, as shown in Figure 3c. The mean value of the coefficient of variation is 0.0999, which is high in the northwest and high in the

southeast. As can be seen from the figure, the stability of the mean AOD in the arid zone of Central Asia during the 20 years is inversely related to the spatial distribution of aerosols, and the regions with high aerosols are weakly stabilized and fluctuate more violently. The central region of the arid zone is sparsely populated and the coercion to the atmospheric environment is weaker, thus the AOD performance is more stable; in the higher altitude region, the AOD continues to be stable; the less stable region mainly lies in the northern part of the Tarim Basin in Xinjiang, China, due to the basin being located in a low-lying area with poor stability; the second is the eastern part of the Aral Sea, which is due to the lack of governance and the degradation of the Aral Sea in the past 50 years. The degradation of the Aral Sea has received worldwide attention, and the wind erosion of the Aral Sea lake floor has resulted in the formation of sand and dust containing a large amount of high-density saline and alkaline dust as well as heavy metals, posing a great threat to the surrounding ecosystem; the entire territory of Tajikistan is included due to the rapid development of the mining industry, and an exciting development in Tajikistan is the planned construction of dams that will make the country rich in hydroelectric energy; and lastly, the northern part of Kazakhstan is included because Kazakhstan has a high GDP compared to the other Central Asian countries, high atmospheric emission capacity, a larger population, and more socio-economic activity.

The Hurst index calculation shows that the range of Hurst index values for the mean value of AOD within the arid zone of Central Asia is between 0 and 1, with a mean value of 0.43. To describe in detail the future trend of AOD in the region, the range of Hurst values is categorized as 0.0–0.25 for strong anticontinuity, 0.26–0.50 for weak anticontinuity, 0.51–0.75 for weak persistence, 0.76–1.00 for strong persistence. From Figure 3d, it can be seen that the persistence characteristics of the arid zone are dominated by weak antipersistence, accounting for 76.26% of the study area, mainly due to the influence of topography and geomorphology and climate. The trend of future AOD changes is opposite to the trend of past AOD changes, and combined with the overall trend, the AOD changes in the area where the AOD has remained unchanged for two decades. Strong inverse persistence is mainly distributed in the basin and lower-elevation areas, accounting for 1.35% of the study area. Weak antipersistence and weak persistence are mosaicked in the western part of the arid zone in winter, and weak persistence is mainly distributed in the region of more extensive human activities, accounting for 22.17% of the study area. Strong persistence is mainly distributed in the area around the Aral Sea, accounting for 0.22% of the study area, indicating that the future trend of AOD change is consistent with the past trend of AOD change.

### 3.3. Interannual Variability of Aerosols

The interannual changes in AOD in the arid zone of Central Asia from 2001 to 2020 are shown in Figure 4, and the annual mean values of AOD in the arid zone in the past 20 years ranged from 0.1416 to 0.1622, with the maximum value occurring in 2006 and the minimum value in 2005. The distribution of the mean AOD values over the past 20 years is relatively stable, with small alternating variations, but the overall trend of the mean AOD values is decreasing; among them, the highest AOD values are found in the Xinjiang region of China, while the lowest mean AOD values are found in Tajikistan. As a whole, the regions with high values are located in Xinjiang, China, and the Aral Sea, where the Tarim Basin has an arid climate, sparse vegetation, and high sand and dust aerosol content. The degradation of the Aral Sea has caused the AOD around the Aral Sea region to increase yearly. The increase in AOD in the vast plains of the study area is mainly due to aerosols emitted from human activities and the effect of biomass burning. In Xinjiang, the decrease in AOD from 2007 is due to the implementation of strict environmental protection policies in China to improve energy utilization. The arid zone of Central Asia is dominated by natural aerosols, with the main sources of dust being in the Aral Sea and the Tarim Basin, which are carried downwind by the wind.

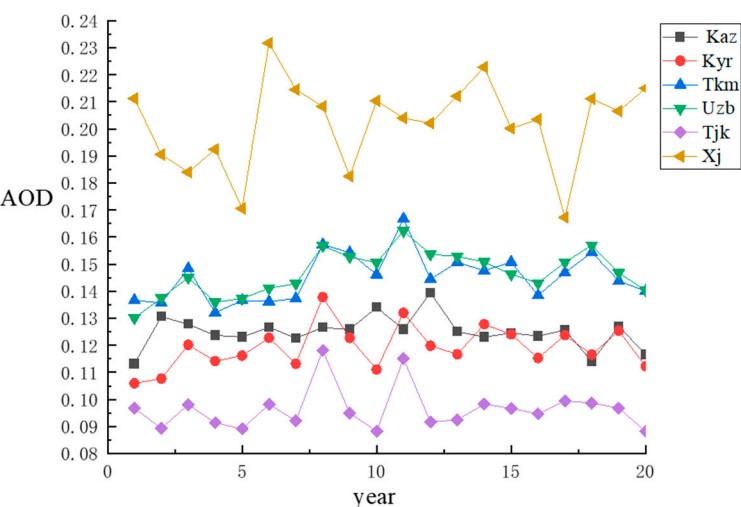

**Figure 4.** Interannual changes in AOD in the Central Asian dry zone, 2001–2020.

### 3.4. Seasonal Changes in Aerosols

As shown in Figure 5, the distribution of mean AOD values in different seasons in the Central Asian arid zone during the 20 years showed obvious differences, and the mean AOD values in spring and summer were significantly higher than those in fall and winter. Regarding the change in seasons, the overall trend showed a decreasing trend: spring (0.1923) > summer (0.1711) > autumn (0.1254) > winter (0.1177), according to the results of related studies. This trend is mainly related to the special geographical location, climate, and topography of the area. The climate of the Central Asian arid zone is arid, the dust is dry and loose, and the monsoon winds blowing from the Atlantic Ocean make it dusty, which makes the frequency of dusty weather in spring and summer significantly higher than that in the fall and winter seasons, which makes the AOD values in the spring and summer seasons higher and indicates that an important factor of the overall high AOD in the Central Asian region is the influence of sand and dust. In the fall and winter, abundant snowfall in Central Asia prevents the local loose dust-prone subsurface from raising dust, and at the same time, the subsurface covered by snow in the fall and winter causes an increase in surface reflectivity, which leads to the overall low AOD in winter. The difference between the AOD values in fall and winter is relatively small, while the distribution map shows that the area of high values increases in winter and is concentrated in human activity areas, which indicates that the high use of fossil fuels caused by human heating activities in winter emits more anthropogenic aerosols, exacerbates atmospheric pollution, and affects the local AOD values to a certain extent.

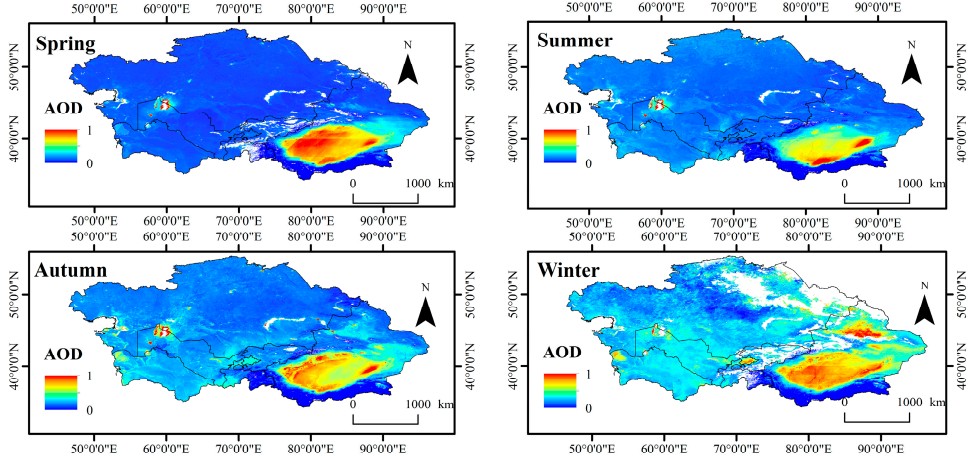

**Figure 5.** Seasonal distribution of AOD in the Central Asian arid zone.

### 3.5. Relationships between AOD, Wind Speed, and Vegetation Indices

This study evaluated the spatial correlation of AOD with wind speed and NDVI in the arid zone. Wind is the nemesis of atmospheric particulate matter, and the magnitude of wind speed directly affects the dispersion of air pollutants. Higher wind speeds may result in wider distribution and dilution of aerosols, reducing their concentration in specific areas and thus AOD values. As shown in Figure 6(a1,a2), most of the correlations between AOD and wind speed were positive, increased diffusion of aerosols prevented accumulation in the study area, with 53.34% of the area showing a positive correlation, the link between optical properties and wind-generated aerosols was not easy to detect, and the correlation coefficients increased with the increase in wind speed [32], consistent with the findings of related studies. The effect of wind speed on AOD values is also interfered with and influenced by other factors such as meteorological conditions, aerosol sources, and atmospheric stability. NDVI affects the ground surface, and when NDVI is low, the ground surface is exposed significantly, which can increase the regional dusty weather [33]. As shown in Figure 6(b1,b2), the correlation between AOD and NDVI is mostly negative. It indicates that AOD is larger in areas with low vegetation cover and small in areas with high vegetation cover, and the reason for this phenomenon is the high level of human activities in areas with low NDVI values; 65.53% of the study area has a negative correlation between AOD and NDVI, which is mainly concentrated around the Aral Sea and in the Tarim Basin. The quality of the ecological environment around the Aral Sea has declined due to poor governance and irrational use of water resources, and the Aral Sea has been shrinking in the past 50 years [34]. For the relationship between AOD, wind speed, and vegetation index, we can find that planting more vegetation in the arid zone not only prevents wind and sand but also improves the vegetation cover.

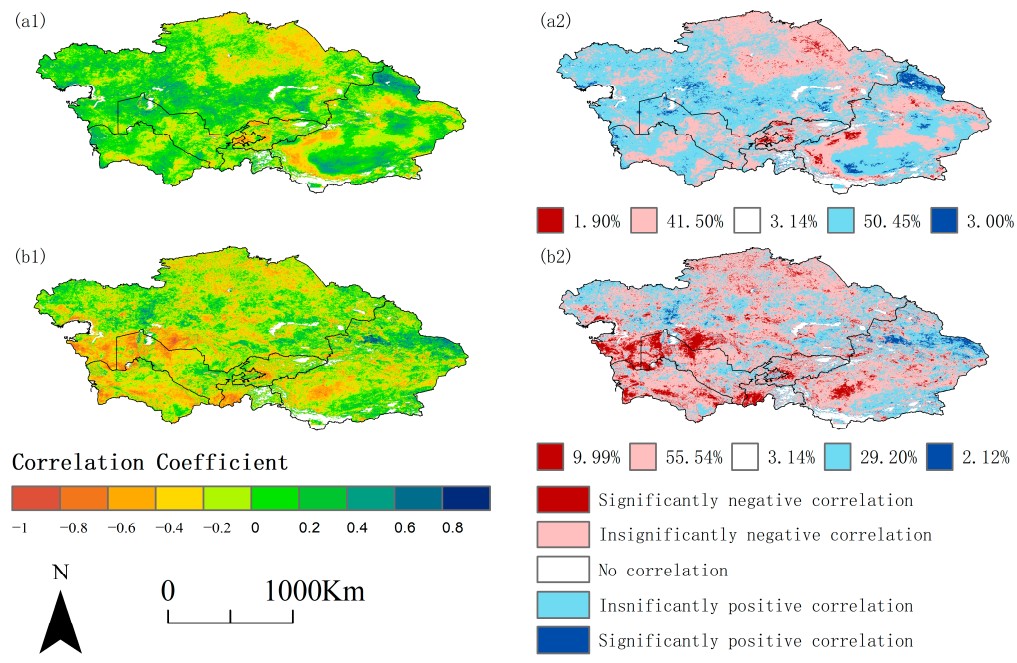

**Figure 6.** Spatial correlation between AOD and wind speed (**a1**) and NDVI (**b1**). (**a2,b2**) indicate the distribution of significant correlations between AOD and wind speed and NDVI.

## 4. Discussion

In this paper, we use MCD19A2 data in combination with AERONET data to validate the adaptability of MCD19A2 in the arid zone, and the main contributions of this study can be summarized as follows: firstly, it is verified that MCD19A2 is well adapted in most of the region and may be a little bit less well adapted at the edge of rivers due to salinization. Secondly, the long-term trend, stability, future trend, and seasonal distribution of AOD

were explored, which provides a basis for understanding the optical characteristics of aerosols in the arid zones of Central Asia. Finally, the study reflected the link between AOD, wind speed, and NDVI, which can help manage the study area's environment.

However, our study has some limitations. First, the observation stations are limited, although we have ten observation stations, including five self-measurement stations. We need to add more stations in future studies to verify the general adaptability of MCD19A2 data. Second, in this paper, only wind speed was considered for the study of AOD and meteorological factors; other meteorological factors such as temperature, precipitation, and surface suitability will be examined in subsequent studies.

## 5. Conclusions

In this paper, the spatial and temporal distribution characteristics of long-term AOD values in an arid zone, especially the trend of changes, were investigated using the MCD19A2 dataset. Some valuable patterns are as follows:

(1)  MCD19A2 is verified to have an underestimation phenomenon by the foundation and measured sites and shows an overestimation phenomenon at the lake and in general has good applicability in the arid zone.

(2)  The annual mean value of AOD in the arid zone for 20 years ranges from 0.1416 to 0.1622, with the maximum value occurring in 2006 and the minimum value in 2005. The distribution of the mean value of AOD over the past 20 years has been relatively stable, with a small degree of alteration, but the mean value of AOD has shown an overall decreasing trend, and at the same time, the mean value of AOD in spring and summer is significantly higher than that in fall and winter. With seasonal changes, the overall trend is decreasing: spring (0.1923) > summer (0.1711) > autumn (0.1254) > winter (0.1177). Poor stability is mainly in the Aral Sea and around the Tarim Basin and in areas with high human activity, and the overall differences in arid zones are significant. The characteristics of AOD persistence in China are dominated by weak antipersistence, with weak persistence and weak antipersistence distributed in a mosaic, and strong antipersistence distributed sporadically in arid areas.

(3)  AOD is positively correlated with wind speed and negatively correlated with NDVI, so it is very important to improve the vegetation cover in the study area.

(4)  In this paper, ground-based and satellite remote sensing techniques are integrated to analyze the characteristics of spatial and temporal aerosol distribution and evolutionary patterns in the Central Asian arid zone from 2001 to 2020 and to evaluate the current status of the atmospheric environment in the region. The next step will focus on the impact of aerosols on climate change and discuss in depth the influence of meteorological factors at different scales on local aerosol patterns.

**Author Contributions:** Conceptualization, Z.Z. (Zhengnan Zhu) and Z.Z. (Zhe Zhang); methodology, Z.Z. (Zhengnan Zhu); software, Z.Z. (Zhengnan Zhu), F.L., Z.C., Y.R. and Q.G.; validation, Z.Z. (Zhengnan Zhu) and F.L.; formal analysis, Z.Z. (Zhengnan Zhu); data curation, Z.Z. (Zhengnan Zhu); writing—original draft preparation, Z.Z. (Zhengnan Zhu); writing—review and editing, Z.Z. (Zhengnan Zhu), F.L. and Z.Z. (Zhe Zhang); supervision, Z.Z. (Zhe Zhang); funding acquisition, Z.Z. (Zhe Zhang). All authors have read and agreed to the published version of the manuscript.

**Funding:** This research was funded by National Natural Science Foundation of China (No. 42061066), Open Project of Key Laboratory in Xinjiang Uygur Autonomous Region of China (2023D04066), and the Open Research Fund of Key Laboratory of Digital Earth Science, Institute of Remote Sensing and Digital Earth, Chinese Academy of Sciences (No. 2020LDE002).

**Institutional Review Board Statement:** Not applicable.

**Informed Consent Statement:** Not applicable.

**Data Availability Statement:** All datasets used in this study are available to download by the web links given in the article.

**Acknowledgments:** We gratefully acknowledge the data provided by the AERONET network.

**Conflicts of Interest:** The authors declare no conflict of interest.

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
