# Peer review of "Study on Accuracy Evaluation of MCD19A2 and Spatiotemporal Distribution of AOD in Arid Zones of Central Asia"

_sustainability, doi:10.3390/su151813959_

Round 1

Reviewer 1 Report

The article entitled “Study on Accuracy Evaluation of MCD19A2 and Spatio-Temporal Distribution of AOD in arid zones of Central Asia” aims to evaluate “…verify the accuracy of the AOD product (MCD19A2) of MODIS satellite remote sensing data on a long time scale, and explore the spatial and temporal distribution characteristics of AOD in the Central Asian Arid Zone (CACAZ) during the period of 20a from 2001 to 2020…”.

Formal questions: - the acronyms must be defined at the appropriate time, for example, the acronyms present in the title consider that they must be defined, as to the reader less accustomed to the subject these acronyms do not demonstrate meaning, for example: atmospheric aerosol optical (AOD). - the objectives must be explicit in the text, as it appears discreetly at the end of the first paragraph of the introduction. - in figure 1 to include an articulation with a world map (or of the Asian continent) allowing the reader to locate the study area globally. It is necessary to define the acronyms: Kaz, Xj, Uzb, Tkm, among others. Indicate the unit of measurement which must be meters above sea level. In this figure, do negative altitudes actually occur in the study area? - A greater detailing of the climate in the study area would be opportune due to the correlation between the variables studied and the climate of the region. - In materials and methods talk a little more about the MCD19A2 sensor. Something similar to this: The MCD19A2 Version 6 data product is a Moderate Resolution Imaging Spectroradiometer (MODIS) Terra and Aqua combined Multi-angle Implementation of Atmospheric Correction (MAIAC) Land Aerosol Optical Depth (AOD) gridded Level 2 product produced daily at 1 kilometer (km) pixel resolution. The MCD19A2 product provides the atmospheric properties and view geometry used to calculate the MAIAC Land Surface Bidirectional Reflectance Factor (BRF) or surface reflectance, MCD19A1 product.

The bibliographic review should be expanded to support the results obtained in this project. The location of the measurement points are located in one of the sectors of the study area. Could this imply a difference in the results? Evaluate whether a detailed description of the methods of comparing the values in particular the correlation coefficient is necessary.

Figure 3: The image resolution must be improved to allow the reader to visualize and read the subtitles. As it is presented, the reading of subtitles is not allowed. What is the explanation for the spatial variation of AOC? Is it associated with land use and land cover? The characteristics of the atmosphere (troposphere)? How does wind speed affect AOD values? Explicitly comment out this association?

What is the unit of measure for AOD? Introduce this in the text. The discussion should be expanded and based on the literature that studies the topic. At the end of the article, the question I ask the authors is: what is the social impact of an article of this nature? Having made these adjustments, I will be delighted to carry out a new reading.

The article entitled “Study on Accuracy Evaluation of MCD19A2 and Spatio-Temporal Distribution of AOD in arid zones of Central Asia” aims to evaluate “…verify the accuracy of the AOD product (MCD19A2) of MODIS satellite remote sensing data on a long time scale, and explore the spatial and temporal distribution characteristics of AOD in the Central Asian Arid Zone (CACAZ) during the period of 20a from 2001 to 2020…”.

Formal questions: - the acronyms must be defined at the appropriate time, for example, the acronyms present in the title consider that they must be defined, as to the reader less accustomed to the subject these acronyms do not demonstrate meaning, for example: atmospheric aerosol optical (AOD). - the objectives must be explicit in the text, as it appears discreetly at the end of the first paragraph of the introduction. - in figure 1 to include an articulation with a world map (or of the Asian continent) allowing the reader to locate the study area globally. It is necessary to define the acronyms: Kaz, Xj, Uzb, Tkm, among others. Indicate the unit of measurement which must be meters above sea level. In this figure, do negative altitudes actually occur in the study area? - A greater detailing of the climate in the study area would be opportune due to the correlation between the variables studied and the climate of the region. - In materials and methods talk a little more about the MCD19A2 sensor. Something similar to this: The MCD19A2 Version 6 data product is a Moderate Resolution Imaging Spectroradiometer (MODIS) Terra and Aqua combined Multi-angle Implementation of Atmospheric Correction (MAIAC) Land Aerosol Optical Depth (AOD) gridded Level 2 product produced daily at 1 kilometer (km) pixel resolution. The MCD19A2 product provides the atmospheric properties and view geometry used to calculate the MAIAC Land Surface Bidirectional Reflectance Factor (BRF) or surface reflectance, MCD19A1 product.

The bibliographic review should be expanded to support the results obtained in this project. The location of the measurement points are located in one of the sectors of the study area. Could this imply a difference in the results? Evaluate whether a detailed description of the methods of comparing the values in particular the correlation coefficient is necessary.

Figure 3: The image resolution must be improved to allow the reader to visualize and read the subtitles. As it is presented, the reading of subtitles is not allowed. What is the explanation for the spatial variation of AOC? Is it associated with land use and land cover? The characteristics of the atmosphere (troposphere)? How does wind speed affect AOD values? Explicitly comment out this association?

What is the unit of measure for AOD? Introduce this in the text. The discussion should be expanded and based on the literature that studies the topic. At the end of the article, the question I ask the authors is: what is the social impact of an article of this nature? Having made these adjustments, I will be delighted to carry out a new reading.

Reviewer 2 Report

See an attached pdf file.

I suggest that the authors have a colleague who is fluent in English edit the paper.
